# l-Arginine and COVID-19: An Update

**DOI:** 10.3390/nu13113951

**Published:** 2021-11-05

**Authors:** Ayobami Adebayo, Fahimeh Varzideh, Scott Wilson, Jessica Gambardella, Michael Eacobacci, Stanislovas S. Jankauskas, Kwame Donkor, Urna Kansakar, Valentina Trimarco, Pasquale Mone, Angela Lombardi, Gaetano Santulli

**Affiliations:** 1Department of Medicine, Einstein-Mount Sinai Diabetes Research Center (*ES-DRC*), Fleischer Institute for Diabetes and Metabolism (*FIDAM*), Einstein Institute for Aging Research, Albert Einstein College of Medicine, New York, NY 10461, USA; ayobami.adebayo@einsteinmed.org (A.A.); fahimeh.varzideh@einsteinmed.org (F.V.); scott.wilson@einsteinmed.org (S.W.); jessica.gambardella@einsteinmed.org (J.G.); michael.eacobacci@einsteinmed.org (M.E.); stanislovas.jankauskas@einsteinmed.org (S.S.J.); kwame.donkor@einsteinmed.org (K.D.); urna.kansakar@einsteinmed.org (U.K.); pasquale.mone@einsteinmed.org (P.M.); angela.lombardi@einsteinmed.org (A.L.); 2Department of Molecular Pharmacology, Wilf Family Cardiovascular Research Institute, Einstein Institute for Neuroimmunology and Inflammation, Albert Einstein College of Medicine, New York, NY 10461, USA; 3Department of Neuroscience, “Federico II” University, 80131 Naples, Italy; valentina.trimarco@unina.it; 4Department of Advanced Biomedical Sciences, “Federico II” University and International Translational Research and Medical Education (*ITME*) Consortium, 80100 Naples, Italy

**Keywords:** antiviral therapies, arginine, cytokine storm, coronavirus, COVID-19, endothelium, immune response, immunity, inflammation, nitric oxide, nitrosylation, oxidative stress, ROS, SARS-CoV-2, T cells, viral infections

## Abstract

l-Arginine is involved in many different biological processes and recent reports indicate that it could also play a crucial role in the coronavirus disease 2019 (COVID-19), caused by the severe acute respiratory syndrome coronavirus 2 (SARS-CoV-2). Herein, we present an updated systematic overview of the current evidence on the functional contribution of L-Arginine in COVID-19, describing its actions on endothelial cells and the immune system and discussing its potential as a therapeutic tool, emerged from recent clinical experimentations.

## 1. Introduction

l-Arginine is a semi-essential amino acid involved in numerous biological processes. It is a substrate for different enzymatic reactions and is metabolized using three major known pathways in the body: (1) Arginase metabolizes l-Arginine to l-ornithine, (2) l-Arginine decarboxylase metabolizes l-Arginine to agmatine, and (3) nitric oxide (NO) synthase (NOS) uses l-Arginine to form NO and citrulline [1,2,3,4].

## 2. Functional Role of l-Arginine in NO Formation

l-Arginine is the substrate used for NO production by NOS [5]; due to its ability to cause NO generation, which has been shown to be a major endothelial relaxation factor (able to increase vasodilation and reduce arterial blood pressure [4,6,7,8]), l-Arginine has considerable potential in becoming a tool to tackle cardiovascular issues [9]. For instance, in patients with known endothelial dysfunction, l-Arginine supplementation (6–8 g per day) has been shown to improve endothelial function and ultimately lower blood pressure [9].

Three isoforms of NOS have been identified; two of them (endothelial NOS [10,11] and neuronal NOS [12,13]) are expressed constitutively, while the last one is inducible and is mainly involved in the inflammatory/immune response [14,15,16,17].

In the reaction carried out by NOS, electrons are transferred to heme in the *N*-terminal domain [18,19]. Electrons are taken from nicotinamide adenine dinucleotide phosphate (NADP) using flavin adenine dinucleotide in the C-terminal reductase domain [20]. Once electrons are transferred to the N terminal oxygenase domain, NO and citrulline are formed via l-Arginine oxidation [5,21,22]. For NOS to function properly, there needs to be an ample amount of l-Arginine available for this reaction [23]. In addition, NADP, glutathione, tetrahydrobiopterin, and oxygen are needed for proper functioning [4,24].

A substrate competition occurs between NOS and arginase [25,26]. Although the affinity for l-Arginine in NOS is much higher than arginase, the speed of the reaction allows for substrate concentration. The speed of arginase rection is a thousand times faster than NOS [27]. Since these two enzymes compete for a common substrate, arginase will reduce the amount of L-Arginine available for NOS to use [28,29], ultimately decreasing the amount of NO produced.

## 3. NO: Friend or Foe?

NO is considered a signaling molecule involved in a number of processes, including inflammatory responses [30]. It is also essential in mediating vasodilation and bronchodilation, in addition to regulating neuronal function, signal transmission, and intraocular pressure [31].

NO acts as an antithrombotic and cytoprotective agent that impedes platelet adhesion, smooth muscle cell growth, and expression of adhesion molecules [32]. A reduction in NO levels triggers a dysregulated control of vascular tone as well as increased thickness and adhesiveness of the vascular wall [33]. NO can also prevent endothelial cells from undergoing programmed death [34].

On the other hand, NO has also been shown to have detrimental effects in many diseases. In some of these cases, there is a shift from the endothelial form (eNOS) of NO synthase to the inducible form (iNOS) [35,36]. Cells that are being damaged due to NO production will express nitrotyrosine, confirming damage from active free radicals [35]. Inflammatory cells are also known to produce free radicals that can cause a reaction with NO made from iNOS, forming peroxynitrate [37]. Peroxynitrate can have a pathogenic effect due to its ability to react with many different molecules, including lipids, amino acids, and nucleic bases. Therefore, peroxynitrate can lead to the dysfunction of tissues by causing a modification of the function of target molecules and their structures, including carbohydrates and lipids [33]. In addition, when peryoxynitrate reacts with nucleic bases, there is a break in single-stranded DNA, contributing to cell damage and apoptosis [38]. The production of free radicals is increased as the underlying diseases progress [39]. These findings suggest that NO and oxygen radicals can overpower the cellular defense mechanisms causing oxidative damage and cell death. Whether or not NO has a toxic or protective effect could depend on many factors [40]. 

Focusing on its action on viral infections, NO is known to have either indirect or direct antiviral activity. A direct effect of NO can lead to inhibition of viruses, and in fact, NO is considered one of the earliest antiviral responses of the host [41], whereas indirect effects include the regulation of inflammation and immune response [42]. NO also plays a key role in the generation of oxidized phospholipids [43], which can operate as potent immunomodulatory signals [44]. NO is necessary for the formation of several reactive oxygen and nitrogen species, including peryoxynitrite, dinitrogen trioxide, and nitrogen dioxide, which all can have an antiviral effect. However, these free radicals can also cause oxidative stress that can lead to severe cytotoxic effects. NO is known to act as a pro-apoptotic inducer in some cells or as an anti-apoptotic modulator in other cell types [45]. For instance, NO has been proposed to contribute to the persistence of hepatitis C virus due to its anti-apoptotic effect in hepatocytes [46,47,48]. Because of these aspects, NO can have both positive and negative roles in viral infections [36,49,50]. In summary, NO has antiviral effects that can be very useful from an immunologic standpoint; however, an excess of NO can also lead to cytotoxic effects [49] (Figure 1).

A clinical study conducted on 14 patients during the first outbreak of the severe acute respiratory syndrome coronavirus (SARS-CoV) in 2003 concluded that inhaled NO treatment for severely sick patients with SARS resulted in improvement of arterial oxygenation and allowed noninvasive pressure support to be discontinued [51]. SARS-CoV is a positive-sense RNA virus that has a genome of approximately thirty kilobases. 

There are some structural proteins that are common among all forms of coronaviruses; these proteins include a nucleocapsid, membrane, envelope protein, and spike (S protein) [52,53,54]. The S protein of SARS-CoV interacts with angiotensin-converting enzyme 2 (ACE2) on the host cells; it has two domains: S1, which is used in receptor binding, and S2, involved in membrane fusion [55].

Akerstrom and colleagues demonstrated that NO inhibits certain steps of the SARS-CoV replication cycle in a concentration-dependent manner, although the exact underlying mechanism was not clarified [56]. In a follow-up study, the same authors proposed two specific mechanisms that NO uses to inhibit the replication of SARS-CoV [57]. The first mechanism involved the disruption of palmitoylating the S protein (depalmytoilation). Such disruption affects the ability of the S protein to interact with ACE2. The second mechanism involved NO reducing the amount of viral RNA replication early on in the replication cycle due to an effect of the cysteine proteases encoded by SARS-CoV [57]; indeed, when Vero E6 cells were treated with S-nitroso-*N*-acetylpenicillamine, a significant decrease in viral RNA production was detected three hours post-infection [57].

## 4. Effects of l-Arginine on the Immune System

A large part of a normal immune system depends on the amount of l-Arginine available in the body. Arginase is known to represent an integral part of certain granulocyte subsets, which can be released locally or systematically once there is an immune response. In addition, there is an accumulation of immature myeloid cells that express arginase, which is released when fighting off specific illnesses. These myeloid cells that express arginase can decrease the amount of l-Arginine [58,59,60].

T cell function has been shown to depend on l-Arginine levels [61,62]. A decreased ability of lymphocytes to proliferate has been reported in critically ill septic patients and correlated to reduced availability of l-Arginine [63]. Moreover, l-Arginine administration has been found to be beneficial to maintain immune homeostasis (Figure 2), especially in terms of T cell and macrophage function [64]. In fact, l-Arginine is essential in the macrophage M1-to-M2 switch [3].

A deficiency in l-Arginine has been shown to lead to a reduction in T cell proliferation and to cause a diminished response in T cell-mediated memory [65]. In vitro assays have validated that L-Arginine can restore the function of T cells [66]. Mechanistically, the immunosuppressive effects of myeloid-derived suppressor cells (MDSCs) due to l-Arginine depletion and lymphocyte mitochondrial dysfunction have been demonstrated in models of cancer [61].

The expansion of MDSCs observed in COVID-19 has been directly correlated to enhanced arginase activity and lymphopenia [67]. Monocytic MDSCs were significantly expanded in the blood of COVID-19 patients and were strongly associated with disease severity; MDSCs were shown to suppress T cell proliferation and IFNγ production, at least in part through an arginase-dependent mechanism, strongly indicating a role for these cells in the dysregulated COVID-19 immune response [68]. Indeed, MDSCs express high levels of arginase, which metabolizes l-Arginine to ornithine and urea, effectively depleting this amino acid from the microenvironment [69]. l-Arginine depletion is known to inhibit T cell receptor signaling, eventually resulting in T cell dysfunction [70] and to increase the generation of reactive oxygen species (ROS), thereby exacerbating inflammation [69,71].

In a recent study focused on COVID-19, Dr. Claudia Morris and colleagues were able to determine the bioavailability of l-Arginine in three cohorts: asymptomatic healthy adults, adults hospitalized with COVID-19, and children hospitalized with COVID-19; they found that both adults and children affected by COVID-19 display significantly lower levels of plasma l-Arginine (as well as l-Arginine bioavailability) compared to controls [72]. Additionally, a low l-Arginine-to-ornithine ratio observed in COVID-19 patients [72] indicates an elevation of arginase activity in these patients. In another study, plasmatic L-Arginine levels were shown to inversely correlate with the severity of COVID-19 [73]. This study also revealed that the expression of the activated GPIIb/IIIa complex (PAC-1), known to be involved in platelet activation and thromboembolic events [74], is higher on platelets of patients with severe COVID-19 compared to healthy controls and inversely correlated with the plasmatic concentration of l-Arginine [73].

These pieces of evidence seem to go against the recently proposed strategy of l-Arginine depletion in COVID-19, based on the assumption that some steps in the viral lifecycle of SARS-CoV-2 could depend on l-Arginine residues (for instance, the nucleocapsid protein has a 6.9% l-Arginine content) [75].

In fact, a decrease in the bioavailability of l-Arginine has been shown to cause a diminished T cell response and function, eventually leading to an increased susceptibility to infections [76,77]. Twelve weeks of continuous l-Arginine supplementation significantly decreased the level of IL-21 [78], while NO has been shown to suppress the proliferation and function of human Th17 cells [79], which have been implied in the pathogenesis of the cytokine storm and of hyperinflammatory phenomena observed in COVID-19 patients [80,81,82,83]. Higher l-Arginine levels are associated with lower levels of CCL-20, a ligand for CCR6, a part of the chemotaxis system that is induced in response to coronavirus infections [81].

In vitro assays have demonstrated that the proliferative capacity of T cells is significantly reduced in COVID-19 patients and can be restored through l-Arginine supplementation [67]. Corroborating these findings, recent metabolomics data indicates that l-Arginine pathways are altered in COVID-19 patients [84] and an increased mRNA expression of arginase has also been found in the peripheral blood mononuclear cells (PBMCs) of COVID-19 patients [85].

Of note, circulating levels of metabolites of the l-Arginine pathway can be affected by arginase activity in red blood cells [86], which is known to be affected by oxidative stress and can contribute to endothelial dysfunction observed in COVID-19 [87]; furthermore, l-Arginine metabolism is known to be altered in hemolysis [88]. The exquisite balance between arginase and NOS activity has also been shown to influence the inflammatory responses of gut resident macrophages [3].

To actually test l-Arginine in COVID-19 patients, based on the rationale described above, we designed a randomized clinical trial to study the effects of adding l-Arginine orally (Bioarginina^®^, 1.66 g twice per day) to standard therapy in patients hospitalized for COVID-19. The interim results, recently published [89], revealed that patients who received l-Arginine had a significantly reduced duration of the in-hospital stay, and a diminished respiratory support, compared to patients in the placebo arm. 

We speculate that l-Arginine supplementation could be also beneficial in controlling long-COVID-19, since the persistence of chronic inflammation and endothelial dysfunction has been shown to be fundamental in COVID-19 sequelae [90,91,92,93].

## 5. l-Arginine Deficiency in African Americans: Implications in COVID-19

The deficiency in l-Arginine could be one of the reasons why African Americans suffer more from cardiovascular disease than other races [94,95,96,97]. For instance, a study completed by Glyn and collaborators compared the l-Arginine profile of African and Caucasian men of similar ages and cardiovascular risk factors. What they found was that levels of l-Arginine were significantly lower in African men while blood pressure and pulse wave velocity were higher [98]. In this study, African American men typically presented with extremely detrimental cardiovascular factors. A total of 292 men (130 of them being African and 162 of them being Caucasian) were studied; in African men, the average level of l-Arginine measured was 107 ± 25.6 μmol/L, whereas Caucasian men were found to have an average l-Arginine level of 126 ± 32.8 μmol/L (*p* < 0.001). However, the authors were not able to determine whether or not the differences in l-Arginine levels were environmentally induced or imposed genetically [98]. 

The deficiency in l-Arginine and l-Arginine derived NO [99,100] could also explain the differences reported in terms of COVID-19 and race [101,102]. In support of this view, the intracoronary infusion of l-Arginine was recently found to have a greater effect on endothelium-dependent vascular relaxation in African Americans than in Caucasian subjects [103].

## 6. Conclusions and Perspectives

The functional contribution of l-Arginine in many biological processes is extremely significant, especially in the control of endothelial and immune activities. There is a strong rationale indicating a beneficial effect of l-Arginine in COVID-19, and preliminary results from a randomized clinical trial seem to support this view.

## Figures and Tables

**Figure 1 nutrients-13-03951-f001:**
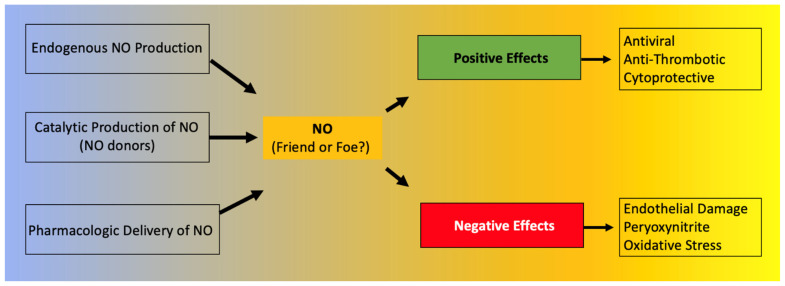
Positive and negative effects of nitric oxide (NO).

**Figure 2 nutrients-13-03951-f002:**
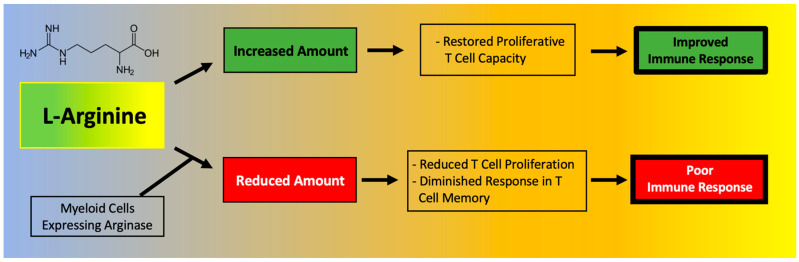
Main effects of l-Arginine on the immune system.

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
