# Peer review of "l-Arginine and COVID-19: An Update"

_nutrients, 2021, doi:10.3390/nu13113951_

Round 1

Reviewer 1 Report

Manuscript ID: nutrients-1447535

Title: L-Arginine and COVID-19: An Update

In the present manuscript, the authors overviewed the relation between L-arginine and COVID-19. This is a very interesting issue for readers working in the field of finding novel nutrients for mitigation of the COVID, only minor issues need to be elaborated before acceptance.

Minor comments:

As the authors indicated, L-arginine and NO have been shown to be a critical role in improving endothelial function and lowering blood pressure. NO has been indicated by several studies for regulating inflammation and immune system, the authors well-reviewed NO’s function on the regulation of inflammation and immune response, it will be helpful if these points could be further linked to L-arginine’s effect on the immune system, for example, L-arginine has been indicated that affected T-cell function directly, has it been correlated to T-cell activation through NO production pathway?

Author Response

Addressed in lines 156-165.

Reviewer 2 Report

This manuscript is a well-written article. The current review contains many complex mechanisms which might be difficult for the readers to understand. A Figure is Worth a Thousand Words. A figure is suggested for depiction of “NO: Friend or Foe?” Similarly, a figure is suggested for depiction of “Effects of L-Arginine on the immune system.”

Minor concerns:

Line 76-77  Whether or not NO has a toxic or protective effect depends on “their diffusion distances”.

Please explain this point in detail and check the reference.

Line 109-111  Please explain the second mechanism in detail to provide the reader a clear concept.

Line 191, 192  The deficiency in L-Arginine and L-Arginine derived NO could explain the differences reported in terms of COVID-19 and race. It is critical to provide the data or hypothesis why levels of L-Arginine were significantly lower in African men or some patients. Consequently, preliminary clinical trial may be performed to support this viewpoint.  

Author Response

Minor concerns:

Line 76-77  Whether or not NO has a toxic or protective effect depends on “their diffusion distances”.

Please explain this point in detail and check the reference.

Line 109-111  Please explain the second mechanism in detail to provide the reader a clear concept.

Line 191, 192  The deficiency in L-Arginine and L-Arginine derived NO could explain the differences reported in terms of COVID-19 and race. It is critical to provide the data or hypothesis why levels of L-Arginine were significantly lower in African men or some patients. Consequently, preliminary clinical trial may be performed to support this viewpoint.